# Assessing the Vulnerability of Agricultural Systems to Drought in Kyrgyzstan

**Li Liang** [1,2], **Fan Zhang** [1,*] and **Keyu Qin** [3]

1   Key Laboratory of Land Surface Pattern and Simulation, Institute of Geographic Sciences and Natural Resources Research, Chinese Academy of Sciences, Beijing 100101, China; Liangl.16s@igsnrr.ac.cn
2   University of Chinese Academy of Sciences, Beijing 100149, China
3   Key Laboratory of Ecosystem Network Observation and Modeling, Institute of Geographic Sciences and Natural Resources Research, Chinese Academy of Sciences, Beijing 100101, China; qingkeyu@igsnrr.ac.cn
*   Correspondence: zhangf.ccap@igsnrr.ac.cn

**Abstract:** As climate change worsens, the frequent occurrence of extreme drought events will further threaten the agricultural systems of all countries in the world. Kyrgyzstan is a country with agriculture and animal husbandry as its main industries, with a weak industrial base, and agriculture plays an important role in the national economy. Kyrgyzstan is located in Central Asia and suffers from a dry climate and frequent droughts. Thus, an integral analysis of the vulnerability of Kyrgyzstan's agricultural system is of great significance for this country's socio-economic stability. In this study, we comprehensively analyze the agricultural system drought vulnerability of Kyrgyzstan from three dimensions of sensitivity, adaptability and exposure. The results show that the areas of higher vulnerability in Kyrgyzstan's agricultural system are distributed in the eastern mountainous, northwest and southwest areas. In addition, regions with low vulnerability are mainly concentrated in the central area. Kyrgyzstan has abundant water resources, but the supporting infrastructure construction is relatively backward. The imperfect irrigation facilities have greatly restricted the development of agriculture and have also increased the vulnerability of the agricultural systems. In the face of climate change, the region may face more severe drought disasters, so increasing infrastructure investment and building a complete irrigation system and water use plan are the keys to reducing the vulnerability of Kyrgyzstan's agricultural system.

**Keywords:** vulnerability; drought; Kyrgyzstan; agricultural system; remote sensing

## 1. Introduction

Global warming is intensifying, and changes in temperature and rainfall have caused extreme drought events in various regions of the world [1]. In recent years, Southwest China [2], the Caribbean, North Africa and India have suffered severe drought disasters [3]. Various signs indicate that global extreme drought not only increased in frequency, but also significantly increased in drought intensity and duration. A long-term severe drought will have a severe impact on the local economy [4], agriculture [5] and human health [6]. Many studies have shown that the impact of drought on the social economy is greater than the sum of losses caused by other natural disasters. It is worth noting that due to the fragile resource environment and backward economic conditions, drought often has a more serious impact on developing countries. In underdeveloped regions, the local residents lack measures to adapt to drought can have a serious negative impact on local livelihoods and health. Therefore, it is of great significance to fully understand and analyze the agricultural system vulnerability to drought in economically underdeveloped areas.

Central Asia is located in the hinterland of Eurasia, where temperature changes dramatically, and due to the Tian Shan Mountains and Pamir Mountains cutting off water vapor transmission from the Pacific Ocean and Indian Ocean, Central Asia has become one of the driest regions in the world [7], most areas belonging to a temperate continental

arid climate [8,9]. Under the influence of climate change, the drought problem faced by Central Asia has become more and more serious. In terms of temperature, the overall temperature in Central Asia has been rising year by year, and the increase rate is much higher than the average level in the Northern Hemisphere [7]. In terms of precipitation, the average annual precipitation is increasingly variable, with a decreasing mean value [10], which leads to an increasing frequency of extreme drought events [11]. In addition, parts of Central Asia are at high altitudes, and water resources are mainly replenished by glacier and snow melt, which makes water resources in Central Asia very sensitive to climate change [12]. Many studies have found that river runoff in Central Asia has been reduced to varying degrees [13,14]. Drought leads to an increase in soil water stress, resulting in an increase in potential evapotranspiration and a decrease in the productivity of plant communities [15], which can easily lead to secondary disasters [16] (such as soil salinization and sandstorms) and large-scale vegetation degradation [17]. Up to now, this kind of vegetation and land degradation and loss caused by drought has appeared in many countries in Central Asia [18,19], seriously threatening the livelihood of local residents and the health of the ecological environment. Therefore, in Central Asia, it is urgent to carry out vulnerability analysis against drought disasters from the aspects of social economy, agricultural production and ecological environment.

Drought vulnerability refers to the disturbance of many factors in the natural system and socio-economic system by drought, which affects the ability of the man–earth coupling system to cope with drought and adversely affects the ecological environment, residents' livelihood and social development [20]. At present, many studies on drought vulnerability have been carried out, many of which are based on the sensitivity-exposure-adaptability research framework provided by the IPCC [21]. Based on this vulnerability analysis framework, researchers have adopted many different methods to analyze drought vulnerability [22]. For example, in order to study the drought vulnerability of peasant households, field visits were conducted to obtain survey data, and corresponding index systems were constructed to analyze vulnerability to drought [23]: using meteorological and remote sensing data combined with the GIS method to study drought vulnerability at the regional scale [24]; fuzzy logic model and multi-criteria decision-making (MCDM) model were used to evaluate drought vulnerability [25,26]; and there have been some studies borrowing machine learning algorithms to carry out drought vulnerability research [27]. In general, the scientific and flexible research framework of sensitivity-exposure-adaptability has been widely used in vulnerability assessment research, and the research results also prove that this is an efficient and effective research method.

Kyrgyzstan is located in Central Asia, bordering China, Kazakhstan and Tajikistan, and it is a typical landlocked country. Since Kyrgyzstan is part of the arid continental climate zone, temperature changes are more sensitive than changes in water, which will accelerate the process of aridization in this region [28], and a continuing drought in Kyrgyzstan will pose a serious threat to agriculture [29]. Kyrgyzstan's agriculture occupies a decisive position in the national economy. More than 62% of people are engaged in agricultural production and agriculture service work, and agriculture accounts for one-third of the gross domestic product (GDP) in Kyrgyzstan [30]. Therefore, the stability and prosperity of agricultural production are of great significance to this country. Due to the observed changes in planting patterns, irrigation development and climate change in recent years, it is necessary to re-examine the country's agricultural vulnerability status and make targeted recommendations based on this current agricultural vulnerability status, to mitigate agricultural vulnerability and improve agricultural and animal husbandry production efficiency [31]. In order to fill the research gaps in this area, this study integrated remote sensing observation data and macro-statistical data and used the sensitivity-exposure-adaptability vulnerability assessment framework to assess the drought vulnerability of Kyrgyzstan's agricultural system on a grid scale. The aim is to have a more comprehensive understanding of the spatial distribution of vulnerabilities in Kyrgyzstan's agricultural system, and to provide reference for local governments and residents to make decisions.

## 2. Materials and Methods

### 2.1. Study Area

Kyrgyzstan, located in the Tian Shan Mountains and Pamir-Alay Mountains in central Eurasia (Figure 1), has a temperate continental climate, strong solar radiation and uneven precipitation distribution, with little summer precipitation, more winter precipitation, annual precipitation of 200–800 mm and is a typically dry-farming region. Land area of Kyrgyzstan is about 119 thousand square kilometers, includes 6 states, 2 cities, 44 districts, 31 cities, 453 prefectures and 1855 towns. The population density averages 30 people per square kilometer. According to statistics, at the end of 2015, the population was approximately 6.362 million. More than a third (33.7%) of the population lived in cities, and about two-thirds (66.3%) of the population lived in rural areas. Affected by transportation, technology and economy, Kyrgyzstan's industry is lagging behind, and agriculture is its main economic pillar. Kyrgyzstan is rich in water resources. However, due to an imperfect irrigation system, the utilization rate of agricultural water is very low, which still faces the problem of water shortage. Climate change also has adverse effects on agricultural production. The monitoring of 19 weather stations in Kyrgyzstan over the past 30 years shows that the temperature in the piedmont region of the Kyrgyz mountains and the Talas valley has increased by an average of 0.05 °C per year. It affects the adaptability and yield quality of crops, as well as their resistance to diseases, insect pests and weeds.

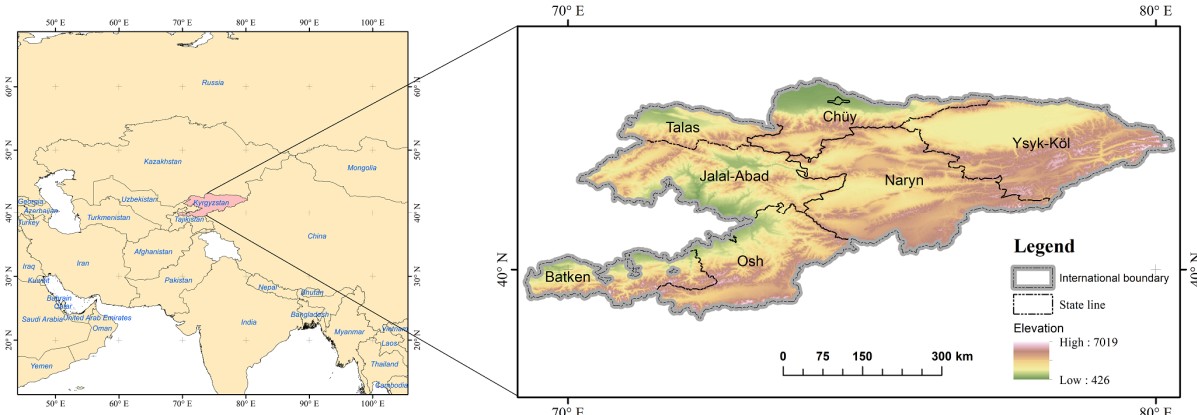

**Figure 1.** Geographical location and topography of study area.

### 2.2. Data Source

The agricultural output of Kyrgyzstan's production and infrastructure investment comes from the National Statistical Committee of the Kyrgyz Republic (Available online: http://www.stat.kg/en/statistics/selskoe-hozyajstvo/, accessed on: 9 October 2020) and Kyrgyz administrative regionalization vector data from GADM (Available online: https://gadm.org/download_country.html, accessed on: 21 October 2020). In addition, the vegetation index comes from the MODIS vegetation index product MOD13A2 (Available online: https://lpdaac.usgs.gov/products/mod13a2v006/, accessed on: 18 April 2021) and the surface temperature data comes from the MODIS surface temperature product MOD11A2 (Available online: https://lpdaac.usgs.gov/products/mod11a2v006/, accessed on: 18 April 2021). Its surface temperature has undergone radiation calibration, cloud removal processing, atmospheric temperature and water vapor correction, and it is calculated by a split window algorithm that establishes a linear combination of 31 and 32 channels of brightness temperature. The population and GDP distribution raster data come from the 2015 data released by the World Bank (Available online: https://data.worldbank.org/country/kyrgyz-republic, accessed on: 20 December 2020).

*2.3. Study Methods*

　　Kyrgyzstan is located in Central Asia and is a typical developing country. Because the basic economy is lagging behind, the statistical data of the country are very rough. This study uses existing data combined with various GIS and remote sensing methods. Exposure, sensitivity and adaptability are three aspects used to study the vulnerability of Kyrgyzstan's agricultural system.

2.3.1. Drought Sensitivity Index of Agricultural System

　　Sensitivity refers to the degree to which the system is affected by climate-related stimuli, including favorable and unfavorable effects. Stimulus refers to all climate change factors, including average climate conditions, climate variability and the frequency and intensity of extreme events. At present, there are various methods of sensitivity evaluation. At first, it was mostly qualitative research. In recent years, it has been widely carried out based on static or dynamic professional quantitative models and linked with prediction results [32]. Input the output climate scene temperature, precipitation, radiation and other variables into professional quantitative models such as crop models, hydrological balance models, etc. [33], and assuming that the temperature, precipitation, radiation, etc., change a quantification to describe the changes in the output variables of the research system, study the system's sensitivity to climate change. In summary, this article defines sensitivity as the degree of response of major crops to drought conditions. The yield of crops in an area determines the sensitivity to drought conditions. In this study, the sensitivity index is characterized by actual crop yield and expected crop yield.

　　Referring to relevant literature on sensitivity and vulnerability [34,35], crop yield is related to climate change, and yield is extremely sensitive to climate change. In this study, we calculated the main crops (wheat, maize, rice, etc.) in 44 regions of Kyrgyzstan by referring to the calculation method of crop failure index of Simelton [36], on the assumption that fertilizer efficiency, planting system, varieties and management measures were the same, and the sensitivity indicators of the main crops are classified according to the crop failure index. The calculation formula is as Equation (1):

$$R_i = \sum^{n} H_{ij} / E_{ij} \tag{1}$$

where $R_i$ represents crop failure index in city i and higher R value represents lower sensitivity. $H_{ij}$ represents real output value of crops in city i and year j, $E_{ij}$ represents the expected output value of crops, in city i and year j, and n represents the sum of the years involved in the calculation. In this study, production data of major crops from 1990 to 2014 have been de-trended and automatically regressed in the Matlab (2017 version) software to eliminate the impact of technological increase or continuous error reporting, to obtain the expected annual output ($E_{ij}$) in city i, year j.

2.3.2. Drought Adaptability Index of Agricultural System

　　The adaptability of the agricultural system has a great relationship with the level of infrastructure and the input of labor. Therefore, we measure the adaptability of the agricultural system in terms of GDP level, fixed asset investment and population density. Since the fixed asset investment data only have state-level data, in order to identify the investment levels of different regions in a more detailed way, we selected 37 high-GDP cities in Kyrgyzstan, according to their proportion of total GDP, according to how the proportion is divided into each city, and then, the fixed asset investment of each city is used to obtain the fixed asset investment level of different regions by using the anti-distance weighting method.

2.3.3. Drought Exposure Index of Agricultural System

　　Exposure is the degree to which the system is subjected to external natural environment or social and political pressure, and the frequency, duration and intensity of external

pressure on the system. Kyrgyzstan is located in Central Asia and has less precipitation. Except for occasional flash floods in mountain areas, flood disasters rarely occur. Drought is the central risk in the study area, so drought is selected as the exposure factor, and meteorological indicators such as drought index can be used to reflect the temporal change and spatial distribution of climate. Due to the small number of meteorological stations in the area, it is not possible to use the SPEI [37] and PDSI [38,39] indices for drought assessment, so we use TVDI (temperature vegetation normalization index) [40] to detect drought conditions in Kyrgyzstan. TVDI was constructed based on the relationship between vegetation index and surface temperature, which was proposed by Sandholt in 2002 [40]. TVDI can be obtained by only using optical remote sensing data to effectively reflect soil moisture on the surface, so it overcomes the shortcoming of insufficient measured data on the surface of Kyrgyzstan. In addition, TVDI proposed water stress index based on simplified NDVI-Ts feature space, which overcomes the deficiency of soil water monitoring based solely on land surface temperature and vegetation index. At present, many studies have proved its effectiveness in complex terrain conditions [41–43]. In order to avoid the impact of inter-annual changes and extreme events on the agricultural drought vulnerability assessment of Kyrgyzstan, this study calculated monthly TDVI from May to September, between 2010–2015, and used the monthly average of TVDI to characterize the agricultural system drought vulnerability of Kyrgyzstan. The calculation method of TVDI is shown in Equations (2)–(4).

$$\text{TVDI} = (\text{Ts} - \text{Ts}min)/(\text{Ts}max - \text{Ts}min) \tag{2}$$

$$\text{Ts}min = a_1 + b_1 * \text{NDVI} \tag{3}$$

$$\text{Ts}max = a_2 + b_2 * \text{NDVI} \tag{4}$$

where TVDI represents temperature vegetation drought index, TS represents surface temperature and Ts*max* is the highest surface temperature corresponding to NDVI, namely the dry edge. Ts*min* is the lowest surface temperature corresponding to NDVI, namely, the wet edge.

### 2.3.4. Drought Vulnerability Index of Agricultural System

Vulnerability refers to the degree to which the system is susceptible to or responds to the adverse effects of climate change (including climate variability and extremes). It is a function of the characteristics and magnitude and rate, sensitivity and adaptability of climate change to which a system is exposed. According to the vulnerability concept and analysis framework proposed by IPCC, vulnerability is composed of three dimensions: exposure, sensitivity and adaptability. Exposure refers to the nature and extent of a system's exposure to climate change. Sensitivity refers to the degree to which a system is adversely affected by climate change. Adaptability refers to the ability of a production system or region to better adapt to climate change [21].

After obtaining adaptability, sensitivity and exposure, the following Equation (5) was used to calculate vulnerability (Equation (5)):

$$V = Exposure \times S/adaptive\ capacity \tag{5}$$

*V*: Represents vulnerability.
*Exposure*: Represents exposure.
*S*: Represents sensitivity.
*adaptive capacity*: Represents adaptive capacity.

## 3. Results

### 3.1. Drought Sensitivity Analysis

Based on the crop data of 44 regions of Kyrgyzstan from 1990 to 2014, the gray prediction model was used to predict the crops of each region in 2015, and the sensitivity (r) of each region was calculated (Figure 2).

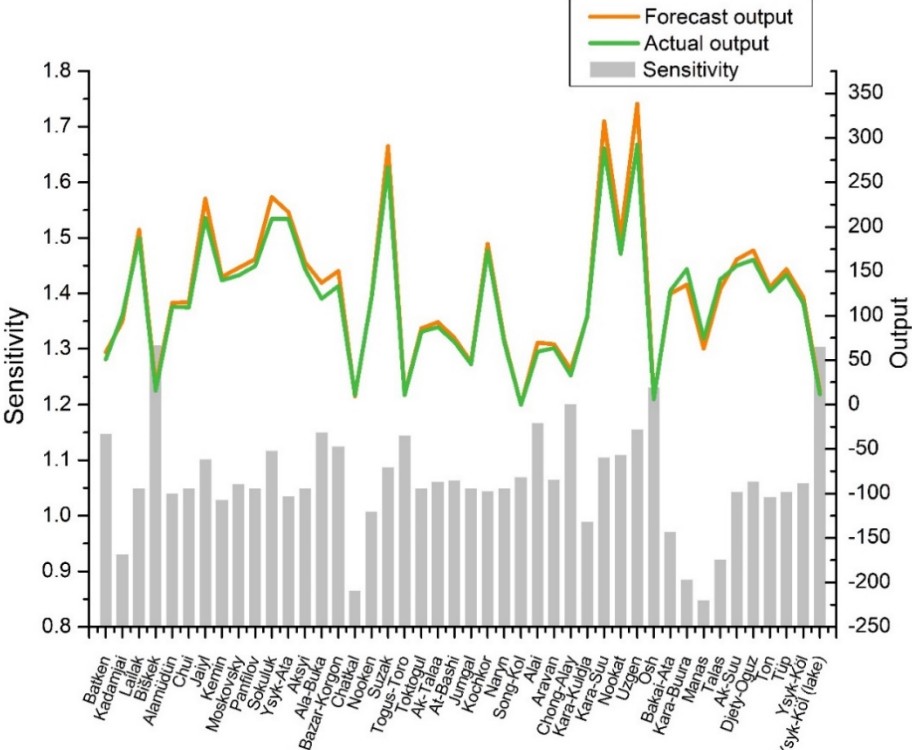

**Figure 2.** Expected crop yield, actual yield and sensitivity of various regions.

As we defined above, agricultural drought sensitivity reflects the extent to which agricultural yields change during drought. The higher value of the sensitivity index, the greater the economic loss that may be suffered when extreme weather events occur. This study calculates the sensitivity of crops in various regions by predicting the ratio of crop yields to actual crop yields. According to the sensitivity analysis results, it can be found that the actual yields in more than 84% of the regions are lower than the expected yields, which indicates that the sensitivity of the crops in the study area is biased. Four areas are classified as extremely sensitive (r ≥ 1.2), ten areas are classified as more sensitive (1.1 ≤ r ≤ 1.2), 23 areas are classified as insensitive (1.0 ≤ r ≤ 1.1), four areas are classified as less sensitive zones (0.9 ≤ r ≤ 1.0) and three areas are classified as extremely insensitive zones (r ≤ 0.9).

The most sensitive is Bishkek, the capital of Kyrgyzstan, with a sensitivity of 1.31, followed by Lake Ysyk-Köl with a sensitivity of 1.31 and then, the southern capital of Osh (Osh) and Uzbekistan. The sensitivity of Chong-Alay is 1.23 and 1.20, respectively; the high sensitivity of agricultural systems in these regions may be due to a combination of various factors, such as the high proportion of main crop area, large average farm size, high rural population density and high proportion of smallholder farmers. Areas with low sensitivity, such as Manas, Chatkal and Kara-Buura, have a sensitivity of 0.85, 0.87 and 0.89 from low to high, respectively. The high sensitivity in these areas may be due to the low sown area of staple food crops, the low proportion of total unirrigated area and the small average farmland size, which makes crop yields more resistant to extreme weather.

It can be seen that the sensitivity of regional agricultural systems in Kyrgyzstan are affected by many factors. First, the scale of agricultural production affects the sensitivity of the agricultural system. The results show that a smaller scale of agricultural production

usually corresponds to higher sensitivity, because even a small amount of yield fluctuation will cause a large percentage fluctuation of the whole, in the case of low crop yield. Second, terrain will also impact sensitivity of the agricultural system. The high-altitude areas present a low sensitivity, and this may be because the high-altitude areas have a small agricultural production scale. On the other hand, it may also be due to the fact that, compared to the low altitude, the high-altitude region has a greater probability of rainfall, so the crops have better production conditions. In addition, social and economic level obviously also have an impact on the sensitivity of an agricultural system. In the case of similar terrain and production scale, the sensitivity of relatively developed areas is relatively low, and this phenomenon is most likely caused by the gap in irrigation infrastructure.

### 3.2. Drought Adaptability Analysis

Adaptive capacity measures the ability of an agricultural system to cope with climate change and its ability to recover from disaster. In this study we assumed that the combination of GDP, population density and fixed asset investment provides the adaptive capacity of Kyrgyzstan's agricultural system. The results are shown in Figure 3 that Kyrgyzstan's agricultural adaptability index has a high spatial distribution in the middle and low distribution in the east and west. Among them, the eastern region, with the highest altitude, has the lowest adaptability, while the western region has lower adaptability, except for the surrounding areas of the city. The agricultural system in central Kyrgyzstan has strong adaptability, especially near the capital city of Bishkek. At the same time, the urban clusters in the central mountainous region also show strong adaptability. Each state of Kyrgyzstan invests in infrastructure construction differently. In the central part, due to the aggregation of several large cities, the fixed asset investment and population density are both high, so it has the highest adaptability, which gradually decreases outward. There are also several relatively developed cities in western Kyrgyzstan, but these cities are not connected together. As a result, only sporadic areas in southern Kyrgyzstan show high adaptability of the agricultural system. Although Kyrgyzstan has abundant water resources, the country still lacks large-scale water storage equipment, which makes it difficult to allocate water resources. At the same time, aging irrigation projects and imperfect supporting facilities in the central and western regions lead to low efficiency in agricultural water use, and limit the expansion of farmland irrigation area, resulting in poor adaptability of agricultural production.

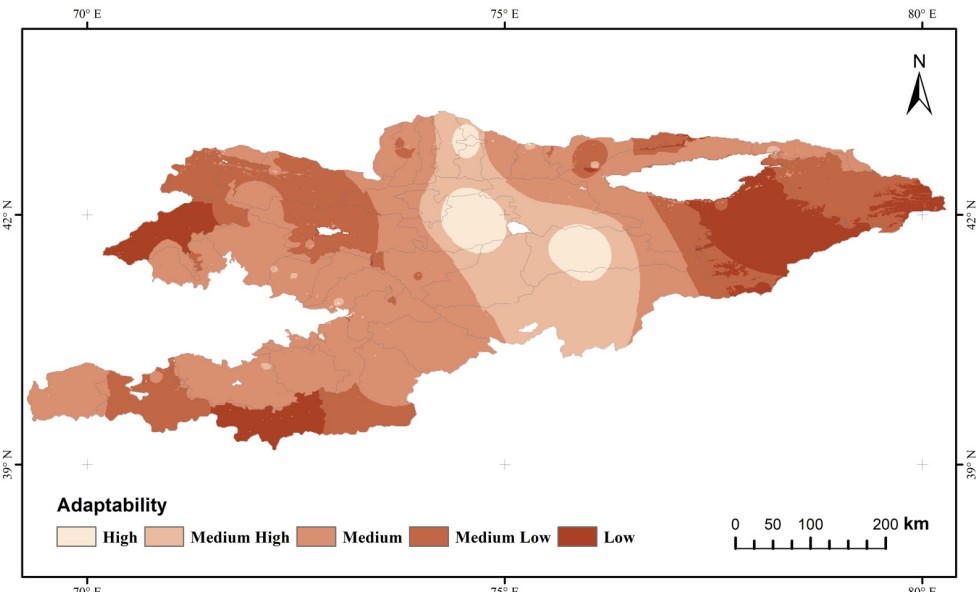

**Figure 3.** Spatial distribution of agricultural system adaptation in Kyrgyzstan.

Although Kyrgyzstan is rich in water resources, due to an imperfect irrigation system, the utilization rate of agricultural water is very low, and this country still faces a water shortage problem. In addition to its own infrastructure defects, climate change will also challenge the ability of Kyrgyzstan's agricultural system to adapt to drought. Research shows that, due to climate change, river runoff in Central Asia will decrease in summer and increase in winter in the future, which indicates the hydrological regime is shifting towards a runoff from snowmelt earlier in the year. Besides, climate change will result in regional precipitation changes, further exacerbating the uneven distribution of water resources, and the increase of glaciers and snow melting will also increase the difficulty of reservoir operation management and water resource dispatching in Kyrgyzstan, which will further reduce the drought adaptability of the agricultural system. In the future, Kyrgyzstan should improve its agricultural adaptability, mainly from two aspects: infrastructure and crop varieties. First, Kyrgyzstan strengthen the construction of irrigation and water conservancy infrastructure, speed up the renovation of irrigation areas, actively promote the reinforcement of reservoirs, improve the guaranteed rate of drought-resistant water sources and establish an efficient agricultural irrigation system. Secondly, the agricultural structure and planting system should be adjusted. Vigorously cultivate and popularize drought-resistant varieties, and actively develop excellent new varieties with great potential for yield increase and wide adaptability.

### 3.3. Drought Exposure Analysis

In this study, drought exposure of an agricultural system refers to the average intensity of regional drought, and the higher the average intensity of drought, the higher the drought exposure, which is represented by the average TVDI value on each grid unit during the dry season from 2010 to 2015. The drought exposure index of the agricultural system in Kyrgyzstan, obtained by using the TVDI index, is shown in Figure 4. The higher the TVDI value is, the drier the region is. On the whole, the spatial characteristics of agricultural system exposure in Kyrgyzstan are high in the west and low in the east, and the exposure index of the high-altitude mountain area is relatively low, while that of the plain area is high. Especially in the mountainous regions of southeastern Kyrgyzstan, the drought exposure index is relatively low, while the flat area at the southwestern border with Tajikistan has a higher drought exposure index. The area around the Toktogul Reservoir also has a higher drought exposure index. It is observed that the spatial distribution characteristics of the drought exposure index in Kyrgyzstan are closely related to the topographical factors. In addition, studies have shown that the spatial distribution of precipitation in Kyrgyzstan is severely uneven. The annual precipitation of the Pskom, Chatkal and Fergana mountains in Kyrgyzstan is 1000~1500 mm, and the rainfall on the northern slopes of the edge mountains such as the Kyrgyz mountains, Taras mountains, Tiexike mountain and Kunge mountain can reach 600–800 mm per year, while the precipitation on the slopes of the inner mountain ranges is only 300–500 mm per year, and the annual precipitation in the Kochikol, Alpa, Alabuga and Naryn basins is only 100~300 mm. It can be seen that the agricultural drought exposure index in Kyrgyzstan has a significant negative correlation with the spatial distribution of precipitation in Kyrgyzstan, which also proves the credibility of the research results, to a certain extent. Overall, terrain and precipitation basically dominate the spatial characteristics of the drought exposure index in Kyrgyzstan. In fact, climate change will further increase the risk of drought exposure in Kyrgyzstan's agricultural system. The monitoring of 19 weather stations in Kyrgyzstan over the past 30 years shows that the temperature in the piedmont region of the Kyrgyz mountains and the Talas valley has increased by an annual increase of 0.05 °C. Although studies suggest that precipitation in Central Asia will increase in the future, the increases in temperature will increase vegetation evapotranspiration [44], leading to higher water requirements for crop irrigation, which will further lead to a crop yield reduction. In order to reduce the risk of drought exposure in the agricultural system, Kyrgyzstan should plan its agricultural space more scientifically [45,46] and reduce the proportion of water-requiring crops planted

in areas with high drought exposure. At the same time, it is necessary to improve residents' understanding of climate change, and improve the ability to forecast extreme climate events and minimize agricultural losses.

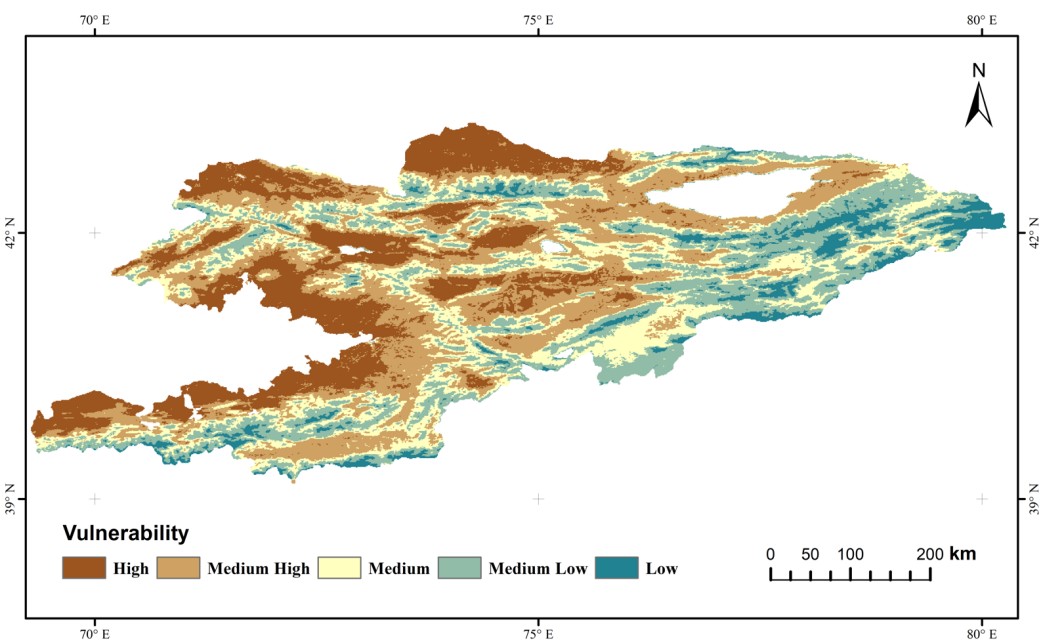

**Figure 4.** Agricultural system drought exposure index map of Kyrgyzstan.

### 3.4. Vulnerability Analysis

By overlaying the three dimensions of sensitivity, exposure and adaptability, the results of Kyrgyzstan's agricultural system vulnerability are shown in Figure 5. Through the above analysis, it is determined that the vulnerability index of central Kyrgyzstan has a high score, and the vulnerability gradually weakens from the central to the east and northwest. The higher-elevation agricultural system in eastern Kyrgyzstan has a higher vulnerability and is not conducive to the development of local agriculture. Although the region has relatively abundant water resources, it is vulnerable to various natural disasters due to the large terrain fluctuations. Agricultural systems are very sensitive and easily affected by factors such as terrain and precipitation. In addition, the infrastructure construction in the region is relatively backward, and the impoundment facilities are imperfect, which results in the weak adaptability of the agricultural system in the eastern region. The above two aspects make the agricultural system in eastern Kyrgyzstan relatively fragile and unsuitable for the development of a planting industry. In addition to the high-altitude areas in the east, the agricultural system vulnerability index in the areas around Naryn and Osh in northwest and southwest Kyrgyzstan is also at a relatively high level. Osh is one of the important agricultural production areas in Kyrgyzstan. However, the results of this study found the region sensitive to climatic factors, it is the agricultural system with a higher degree of drought exposure and the backward agricultural infrastructure makes its adaptability relatively weak. The combination of these factors leads to the relatively high vulnerability of the agricultural system in this region.

The regions of Kyrgyzstan's agricultural system with low vulnerability are mainly distributed in the central region, especially around the capital Bishkek and the central urban areas. Although the agricultural systems in these regions are highly exposed and vulnerable to drought, the infrastructure construction is relatively complete, and the agricultural systems are highly adaptable. At the same time, compared with other regions of Kyrgyzstan, the social economy of Kyrgyzstan is relatively high, and the development of the crop industry is at the leading level in the country, which also makes the crops in the region less sensitive. Generally speaking, although the central region's agricultural system drought vulnerability is low, Kyrgyzstan's agricultural system as a whole is rela-

tively vulnerable. To improve agricultural vulnerability in various regions of Kyrgyzstan, the Kyrgyz government should give priority to the development of water conservancy irrigation infrastructure and integrated water-saving irrigation systems in agricultural areas with a high vulnerability index and extensively carry out economic and technical cooperation in agricultural research, education, training and promotion, to improve the adaptability of agricultural systems and reduce agricultural vulnerability.

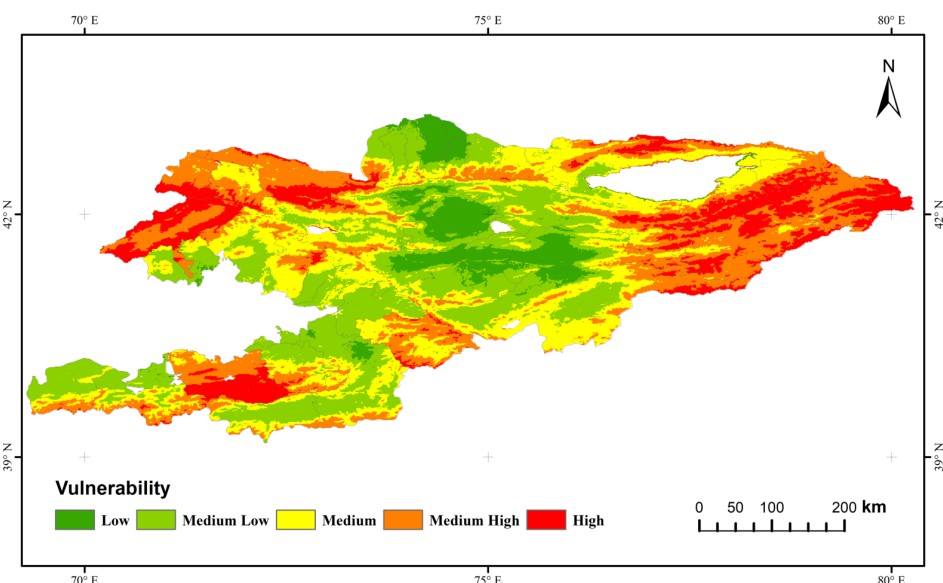

**Figure 5.** Agricultural system drought vulnerability index map of Kyrgyzstan.

## 4. Conclusions

This study analyzes the vulnerability of Kyrgyzstan's agricultural system from three aspects: sensitivity, adaptability and exposure. It is hoped that this study can help local governments make relevant policy decisions and provide references for investors in China and other countries. The results show that the regions with higher vulnerability in Kyrgyzstan's agricultural system are mainly distributed in the southeast, northwest and southwest regions, and the southeastern regions with higher altitude are the most vulnerable. The regions with relatively low vulnerability are distributed in central Kyrgyzstan and the northern region centered on the capital Bishkek. Kyrgyzstan is a mountainous country with arid and semi-arid regions. The unique geographical conditions provide it with abundant water resources, but at the same time, the backward infrastructure makes it impossible to effectively use the water resources in the country, supported by agriculture. Kyrgyzstan urgently needs to establish an infrastructure guarantee system that can guarantee agricultural water use, so as to better face the possible drought impacts caused by climate change.

**Author Contributions:** Conceptualization, F.Z. and L.L.; methodology, F.Z., L.L. and K.Q.; formal analysis, F.Z. and L.L.; data curation, L.L.; writing—original draft preparation, L.L.; writing—review and editing, F.Z. and L.L.; supervision, F.Z. All authors have read and agreed to the published version of the manuscript.

**Funding:** This research was supported by the Strategic Priority Research Program of the Chinese Academy of Sciences (Grant Nos. XDA20010302) and the National Natural Science Foundation of China (Grant No. 72004215).

**Institutional Review Board Statement:** Not applicable.

**Informed Consent Statement:** Not applicable.

**Data Availability Statement:** Not applicable.

**Acknowledgments:** This research was supported by the Strategic Priority Research Program of Chinese Academy of Sciences (Grant Nos. XDA20010302) and the National Natural Science Foundation

of China (Grant No. 72004215). We would also like to thank the reviewers for their thoughtful comments that helped improve the quality of this work.

**Conflicts of Interest:** The authors declare no conflict of interest.

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
