# Peer review of "Assessing the Vulnerability of Agricultural Systems to Drought in Kyrgyzstan"

_water, doi:10.3390/w13213117_

Round 1

Reviewer 1 Report

The paper is about a serious and important topic (agriculture vulnerability in semi-arid areas) however, there is a lack of focus. The quality of the presentation is low. Moreover, it is not described in a satisfactory way the research core of the paper. This affect the presentation of the results, which seem not relevant sometime (Is it needed to made complex evaluation to say this: 'Generally speaking, areas with flat terrain, sufficient irrigation resources, fertile soil 254 and convenient transportation are suitable for crop development, and generally have low 255 sensitivity. On the contrary, some areas with high levels of urbanization and incomplete 256 agricultural systems usually have high crop sensitivity'?). I suggest to make stronger the introduction and the description of the aims of the paper. 

Some spotted observations: 

- Please better explain the data sources (par. 2.2): not all of them are linked

- ln 142: like what? Add specific references

- Please avoid repetitions (e.g.: double definitions of drought sensitivity at begin of 2.3.1 and 3.1) 

- Change 3.1. Sensitivity analysis in 3.1. Drought sensitivity analysis (or index) as Sensitivity analysis without any adjective refers to something different. 

- Last issues: there are several typos (in particular punctuation issue) and some mismatches of the references list

Author Response

Dear reviewer 1,

On behalf of my co-authors, we thank you very much for giving us an opportunity to revise our manuscript. We greatly appreciate your positive and constructive comments and suggestions on our manuscript.

We have studied your comments carefully and made revision in the places marked in red in the paper. First, in view of the problem of low presentation of results, we have made major modifications to the result part, rewriting the last paragraph of section 3.1, and also making major modifications to the expression of results in 3.2 and 3.3. At the same time, the significance and focus of the research are emphasized in the abstract and introduction. Then, we corrected the details you mentioned one by one, indicated the connection of data sources, added corresponding references in 142 lines, corrected the repeated content in the article, changed the statement of section titles, and corrected the incorrect punctuation and typesetting in the article

We have tried our best to revise our manuscript according to the comments. Attached please find the revised version, which we would like to submit for your kind consideration. We would like to express our gratitude again to you for the comments on our paper. I am looking forward to hearing from you.

Yours sincerely,

Liang Li

Name: Zhang Fan    E-mail: zhangf.ccap@igsnrr.ac.cn

Reviewer 2 Report

Major comments:

This manuscript presents a remote-sensing based analysis of agricultural sensitivity to drought and climate change in Kyrgyzstan.  The topic is of interest to readers of Water, and the study is generally well written.

My biggest concern with the study is the with the TVDI.  I think this needs to be explained better.  My concern is that the minimum and maximum temperatures are set spatially (rather than for each pixel over time).  While thermal-based approaches are useful for detecting drought or evapotranspiration, they have strong limitations where there are major changes in elevation.  These changes in elevation introduce variations in temperature that are independent of moisture status.  I am concerned this elevation effect propagates into the results.  For example, there seems to be a very strong correlation between elevation and exposure (Fig. 4). 

Specific comments:

Line 10 and Line 26:  Clean up placeholder text in correspondence and citation section:

Line 52:  I would use “is increasingly variable with a decreasing mean value” instead of “is fluctuating and decreasing”

Line 105:  I would try to condense this paragraph.

Line 123:  Replace “annual” with “average” and add “per year” after 0.05 C to indicate that this is a rate.

Author Response

Dear reviewer 2,

On behalf of my co-authors, we thank you very much for giving us an opportunity to revise our manuscript. We greatly appreciate your positive and constructive comments and suggestions on our manuscript.

We have studied your comments carefully and made revision in the places marked in red in the paper. And we think it is necessary to address your main concern (the effect of topography on surface temperature). In fact, at the beginning of the study, we considered the possible influence of terrain factors on TVDI index and the influence of terrain on surface temperature. However, we considered that TVDI index mainly reflects the degree of regional drought by soil moisture. In higher altitude areas, although the surface temperature is lower, but its atmospheric temperature, atmospheric pressure and saturation vapor pressure is low, which leads to the soil and vegetation evaporation is low, which leads to the increase of soil moisture, so even if the same surface temperature, high altitude of soil moisture is higher than that of low altitude area of soil moisture, consider these comprehensive factors, we did not use DEM modified TVDI index. In fact, several studies have applied TVDI index to mountainous areas and proved that TVDI still has a high accuracy in drought monitoring in mountainous areas.

We have tried our best to revise our manuscript according to the comments. Attached please find the revised version, which we would like to submit for your kind consideration. We would like to express our gratitude again to you for the comments on our paper. I am looking forward to hearing from you.

Yours sincerely,

Liang Li

Corresponding author: Zhang Fan    E-mail: zhangfan.ccap@igsnrr.ac.cn

Reviewer 3 Report

Dear Authors,

The submitted paper is acceptable for publication; however, a few issues need to be discussed and some information needs to be added to the paper. Since the observed increase in air temperature is likely to persist, then it will most likely affect water resources. Thus the first question concerns such impacts. In addition, some hydrologic and meteorologic issues need to be discussed.

- Does the annual precipitation total for previous years follow some observable tendency (increase, no change, decrease)?

- Is the river regime-changing or has it changed in the past? The paper does not provide information on this issue. When does high and low discharge occur?

- How will higher air temperatures affect water resources? Is it possible that the supply of water in rivers will increase along with an increase in air temperatures due to the melting of mountain area glaciers?

Author Response

Dear reviewer 3,

On behalf of my co-authors, we thank you very much for giving us an opportunity to revise our manuscript. We greatly appreciate your positive and constructive comments and suggestions on our manuscript.

We have studied your comments carefully and made revision in the places marked in red in the paper. First, we discussed future changes in rainfall in Kyrgyzstan in section 3.2. Then, we discussed future changes in river runoff in section 3. And when we discussing the above two problems, the factors of temperature rise have been taken into account. The rise of temperature leads to the advance of glacier melting, thus changing the seasonal variation of river runoff, and at the same time improving evapotranspiration of regional vegetation and increasing water demand of vegetation.

We have tried our best to revise our manuscript according to the comments. Attached please find the revised version, which we would like to submit for your kind consideration. We would like to express our gratitude again to you for the comments on our paper. I am looking forward to hearing from you.

Yours sincerely,

Liang Li

Name: Zhang Fan    E-mail: zhangf.ccap@igsnrr.ac.cn

Round 2

Reviewer 1 Report

The authors improved the paper. I have no further questions

Author Response

Thank you very much!

Reviewer 2 Report

The authors have not addressed my concerns with the TVDI analysis and how it is confounded by elevational differences.  While I agree with the authors that relative soil moisture is higher (and evapotranspiration is lower) with increased elevation, the surface temperature differences from elevation alone are going to result in a conclusion that higher elevations are less susceptible to drought.

I think you need to calculate TVDI either using a detrended surface temperature or on a per-pixel basis (preferred).  Calibrating anomalies of TVDI for each individual pixel will highlight vegetation anomalies and their relationship to drought years.  It might be that warmer and drier years see a positive TVDI anomaly at high elevations (longer growing season) and a negative TVDI anomaly at low elevations (decreased water availability and higher heat stress).  However, your current calculation of TVDI makes it impossible to assess water status and vegetation relationships independently of elevation.

Author Response

Dear reviewer 2,

On behalf of my co-authors, we Thank you again for your valuable comments on this article.

We have carefully considered your comments and believe that the problem you mentioned is very important and critical. As you suggested, we recalculated TVDI in the per-pixel manner you mentioned, replacing the original exposure and vulnerability results in the article with the new results. The overall trend of the new results is basically the same as that of the old results, but the drought exposure of the new results is slightly lower than that of the old results at high altitudes.

Attached please find the revised version, which we would like to submit for your kind consideration. I am looking forward to hearing from you.

Yours sincerely,

Liang Li

Corresponding author: Zhang Fan    E-mail: zhangf.ccap@igsnrr.ac.cn

Round 3

Reviewer 2 Report

Thank you for making these changes.  I have no further review comments.